# A Survey: Security, Transparency, and Scalability Issues of NFT’s and Its Marketplaces

**DOI:** 10.3390/s22228833

**Published:** 2022-11-15

**Authors:** Sangam Bhujel, Yogachandran Rahulamathavan

**Affiliations:** Institution of Digital Technology, Loughborough University London, London LE11 3TU, UK

**Keywords:** NFTs, blockchain, marketplace, scalability, transparency, security

## Abstract

Non-fungible Tokens (NFTs) are ownership records stored on a blockchain, and they are typically digital items such as photos and videos. In many ways, an NFT is like a conventional proof-of-purchase document, such as a paper invoice or an electronic receipt. NFTs are attractive among other things because of verifiability; each sale is recorded as a blockchain transaction, allowing ownership to be tracked. Also, NFTs can be used to transfer digital assets between two mutually distrusting parties, since both the crypto payment and the asset transfer take place in one transaction. With NFTs, all marketplaces can freely trade with the help of decentralized applications (DApps). It is currently estimated that there are over 245 NFT marketplaces (NFTM) listed with over 1000 blockchains as of August 2022 with 68 million blockchain wallet users. With the expansion of markets, they must face challenges and issues. The objective of this review is to study the market dynamics of NFT ecosystems. It also focuses on technical components that enable NFTs and their marketplace. The review provides a deeper understanding of its components, how they are related, and why they are important. The paper analyses the challenges faced by NFTs and marketplaces in terms of security, transparency, scalability, and the consequences leading to these issues and how they will address them, as well as future opportunities.

## 1. Introduction

An NFT (non-fungible token) represents ownership of virtually anything, from digital art to music, videos, memes, and virtual real estate. Additionally, the value of each NFT is unique, and they are considered non-fungible. The term NFT originated with the eToro CEO Yoni Assia who coined the term “colored coins” and co-authored a whitepaper entitled, “Colored Coins” with Ethereum creator Viralik Buterin and their associate [1,2] In this system, colored coins are categorized with transactions with additional features such as customizing by adding metadata. By utilizing its framework, one can add information to a transaction and separate one coin from another. However, despite this being a cryptography-based design, its concepts helped other developers understand how a particular coin can represent a real value as a virtual bridge to actual assets (such as properties, stocks, and contracts). 

NFTs play a major role not only in the digital domain but also in linking physical assets (known as physical asset NFTs) in a digital domain through blockchain. Examples include NFTs for real estate [3], diamonds, collectibles, gold, and many other assets [4]. Sensors (Internet of Things) play a major role in bridging physical assets in a digital domain via extracting vital attributes of the physical asset i.e., GPS coordinates for lands, measuring the unique physical characteristics of diamonds, etc. [5].

The first NFT was created in May 2014 by Kevin McCoy and referred to as Quantum on the Namecoin blockchain. Namecoin is a peer-to-peer naming system-based software built from the bitcoin blockchain [6]. Several different experiments with different development occurred following these events, including NFTs within the Ethereum blockchain. Also, because of colored coins, people began to realize that there were a lot of possibilities for issuing assets over the blockchain. Nonetheless, people understood that bitcoin itself, in that incarnation, was not intended to enable such features. During this time, a new platform was created to address those problems. As a result, in 2014, Dermody and his colleague founded Counterparty, a peer-to-peer financial platform and a distributed, open-source internet protocol based on the bitcoin blockchain. In the process, more and more projects and assets were developed, including a trading cards game (2015) and a meme trading game (2016) [7]. Only after 2017 did real NFTs shift to Ethereum with Ethereum blockchain and what constitutes an NFT today. Moreover, it was backed up by the enunciation of a set of token standards, enabling developers to create their own tokens.

The process of recording and processing NFTs can be seen in Figure 1 [8]. Data from a blockchain transaction is digitized into the appropriate format, then stored in an external or internal database based on cost. After signing the transaction, the owner sends the data to the smart contract with a hash. In this case, smart contracts process the data, mint, and trade it on the blockchain as a transaction. Upon validation, the NFT is permanently associated with the unique hash identifier, which is stored in the blockchain records permanently.

### Token Standard

The NFT serves as a means of identifying something or someone uniquely. These tokens are subject to their own standards, among which are the Ethereum Request for Comments 721 [9] and the ERC-1155 [10]. In contrast, ERC-1155 is more like an upgraded version of ERC-721, where ERC-721 is a foundation for NFTs; ERC-1155 mixes fungibles and nonfungibles into one contract with trails for NFTs.

These standards helped to make NFTs, and the characteristics they are known for today. As new industries enter the marketplace, the NFT is finding new ways to provide value across a wide range of digital assets, including videos, collectibles, and digital arts. Figure 1 shows the top 10 NFT collections based on sales volume. This visualization was created based on the HEMIL NFT dataset [11]. Since 25 November 2021, this dataset collection has been updated every 15 days, most recently on 2 October 2022.

According to Figure 2, the most popular NFT collection currently is Axie infinity, which has been sold for $4.07 billion worldwide. Axie INFINITY [12] is an Ethereum-based video game where the player collects and mints NFTs using an electronic pet called Axie. Axie Infinity has a total of 1,758,089 buyers and 2,421,759 owners. Following them are CryptoPunks with $2.27 billion and Bored Ape Yacht club with $2.22 billion.

## 2. Relevant Review Articles and Research Methods

NFTs are new and there are not many review articles in the literature. One survey paper [13] deals with the security aspects of blockchain systems. This study provides in-depth information on all the mechanisms that blockchain provides in addition to a brief overview of blockchain technologies. However, their primary focus is on security issues, including ransomware, DoS attacks, underground markets, etc. In addition, there is a report that provides a general overview of NFTs [14]. Despite its extensive details about the overview and challenges of NFTs, the report is missing some information, such as scalability, the NFT marketplaces, and the underpinning components of NFTs and NFTMs. In addition to survey papers, there are some papers whose research direction is worth mentioning even though they are not particularly survey papers. This paper examines NFTs because they have attracted a great deal of investor interest, with some NFTs being sold at unimaginable prices with unthinkable assets, and they ask about scarcity and value with regard to blockchain technology [8]. The next paper explores the state of the literature on NFTs in the economic and financial domains with published papers to help future researchers see new opportunities in other fields such as asset pricing, to enomics, and Risk and Regulation [15]. The paper investigates the current state of affairs concerning NFTs and opportunities for fundraising for galleries, libraries, archives, and museums (GLAM) [16]. NFTs and their collection are introduced in the paper, as well as what the future holds for the arts (Art n.d.). The final research paper examines NFTs from a business perspective and discusses their marketing implications using the modified AIDA (awareness, desire, action, and recurring action) hierarchy that gives marketing guidelines to persuade consumers to buy NFTs due to their scarcity, non-fungibility, and other characteristics [17]. In contrast, the proposed review provides an in-depth analysis of current NFT ecosystems in terms of security, transparency, and scalability. To the best of our knowledge, there are no other review papers in the literature that match the proposed paper.

### 2.1. Purpose of the Study

The purpose of this study is to survey the NFT and its marketplaces as well as to examine the most critical elements in the NFT ecosystem, which are scalability, transparency, and security. These three elements are essential to the NFT ecosystem and are needed for the system to function correctly and efficiently. The research process involves several qualitative methods which are used to gather the information for this study for the purposes of measuring, categorizing, identifying, and generalizing. All these research data have been gathered from secondary sources, and they can be understood by beginners and will provide a better understanding of the current landscape in the field of NFTs.

### 2.2. Structure of This Review Article

This review article contains seven distinct sections, starting with the introduction and research methods. Thirdly, a literature review on the pillars of NFT mechanisms such as blockchain, smart contracts, and transactions focusing on scalability, transparency, and security issues. The fourth section analyses the top five blockchains influenced by current market conditions. The fifth section introduces the NFT marketplace and analyses its underlying components. The sixth section summarizes the challenges currently being faced in the categories of scalability, transparency, and security. The conclusions and future direction for NFTs are discussed in the final section.

## 3. The Pillars of NFT

The blockchain is a distributed ledger that is shared among nodes in a computer network. Blocks are communicated between nodes by a peer-to-peer network, and a consensus protocol is used to validate new blocks. In essence, blockchains are collections of information that are held in groups known as blocks, which store a variety of information. Each block can hold a certain amount of information and once each block has been filled, it closes and links with the previous block to form a chain.

The first decentralized blockchain proposal was made by Satoshi Nakamoto in 2008, a peer-to-peer electronic cash system [18], and in 2009, the authors created the first cryptocurrency, also known as bitcoin, and recorded the transaction on the genesis block [19] as shown in Figure 3. Blockchains are distributed ledgers that can be accessed by anyone and contain extremely secure encryption algorithms. Once the data is stored in a blockchain, it is very difficult to change the data, as each block contains a hash of the previous and the hash of the block itself. The hash keeps everything connected and secure, so if it changes then the block will no longer be valid. There are different types of blockchains with different types of information stored inside the blocks. For example, Bitcoin stores transaction details, sender, receiver, and coins. A blockchain is a system that also has different layers and each layer is on top of the other with no partial dependency. For example, layer 1 does not depend upon layer 1 but layer 2 depends on layer 1 in some areas. Under a blockchain, layer 0 is the network framework, which includes protocols, connections, hardware, and anything else that gives the blockchain its foundation. Layer 1 is the blockchain itself, which is built upon the layer 0 base protocol. The layer 1 consensus protocol is available in either proof-of-work or proof-of-stake forms; more details can be found in Section 3.4. Layer 1 blockchains, such as bitcoin and Ethereum, sacrifice one of the two founding principles of the blockchain (Security, Scalability, and Decentralization) to achieve layer 1 blockchain status. Bitcoin, for example, is secure and decentralized, but it lacks scalability and speed. As a result, Layer 2 protocols offer an efficient, high-speed, and more efficient solution to this scaling problem. For instance, polygon and immutable X are examples of layer 2 on the Ethereum blockchain. Layer 3 is another layer which is also called the application layer and contains APIs, user interfaces, and smart contracts. On Ethereum, there are a number of DApps, e.g., Dark Forest, Foundation, etc.

A bitcoin uses a Proof of Work (PoW) consensus mechanism, which is the basis for all other blocks in the bitcoin network. Arthur Gervais describes PoW as a way of verifying the accuracy of new transactions on a blockchain [20]. In the early days of cryptocurrency, there was no “next Bitcoin”; however, we eventually witnessed the emergence of a new blockchain called Ethereum, which changed the limitations on the functionality of digital currency.

In terms of market capitalization, Ethereum is the second largest blockchain after Bitcoin [21]. Ethereum is an open-source blockchain platform that anyone can use to create any type of secured digital ledger technology. Although it was founded in 2013 by Vitalik Butlerin, the introductory paper for Ethereum was first documented in 2014 [22]. In it, the developers explain the technology and purpose of the project they are working on. In an interview, Butlerin compares bitcoin to a pocket calculator that is “good at doing one thing,” as opposed to Ethereum, which is more akin to a smartphone with a variety of applications you can use and additional applications that others can create [23].

### 3.1. Smart Contract

A smart contract is a type of transaction protocol. They are self-executing contracts in which the terms of the agreement between buyer and seller are written in code and are automatically verified and executed via a computer network [24]. Nick Szabo proposed it first in the 1990s [25]. A contract that powers decentralized applications is known as a smart contract (Dapps). As it executes and codifies effects such as the transfer of some value between parties, smart contracts can be regarded as a secure storage procedure. The introduction of Bitcoin in 2009 made the intelligent contract a reality. While multi-signature wallets and other smart contracts are possible on bitcoin, the more versatile and complex smart contracts that are widely discussed today are primarily found in Ethereum. The Ethereum protocol was responsible for elevating the technology to the status of a foundational component of the blockchain [26]. The main concepts of smart contracts are deployed into blockchains, which then render transactions traceable, transparent, and irreversible for that block/information while travelling to the next chain of blockchain.

### 3.2. Addresses and Transactions

An address in the context of blockchain typically consists of a string of 26 to 35 characters and is a particularly distinctive identification for an entity, a wallet, or a smart contract and a public ledger known as blockchain is the hub of NFT which handles transactions [27,28]. Another important property is cryptography. When a bitcoin is created on a blockchain, its address is a 160-bit hash of the public key derived from the private key of the Elliptic Curve Digital Signature Algorithm (ECDSA). It takes three steps to create an address: first, create the private key; second, extract the public key from the private key; and third, hash the public key to create the address. This address is a value in a cryptocurrency, and any time a transaction takes place, it’s referred to as an event. This transaction contains the value, message, sender, receiver, timestamp (little data saved in each block with its own unique serial) [29], the block to which the transaction data belongs, and the hash of all the transaction data combined [30].

### 3.3. Speed

Among the biggest and most important aspects of a blockchain is transaction speed. When you need to issue thousands or even millions of NFTs at once, or even trade at once, speed becomes necessary. Transaction speeds are determined by both the complexity of the transaction and the network traffic. The minting [31] process of converting digital files into digital assets stored on the blockchain also requires intensive computations, so gas fees are the way to conceive of assisting miners in maintaining blockchain records. Depending on the current gas price and the demand, a single NFT transfer can cost between 50 cents and $15.00 [32]. Each fee falls into one of three categories:-The first is the gas fee, which is mainly used for storing and transacting NFTs on the blockchain.-The second fee is the account fee, which is charged by the NFT marketplace for each account.-The third charge is the listing fee, which is charged to list a digital asset (NFTs) for sale.

In general, NFT prices differ between blockchains and markets, and are mostly based on what the user wants, such as the amount of data used, the speed, and availability.

### 3.4. Scalability

Each transaction in the system is organized in blocks, which are then linked in a chain. Several metrics can be used to measure scalability, including maximum throughput, latency, bootstrapping, and cost per confirmed transaction (CPCT) [33], where a block is the one being transported. As transaction volumes increase every day, the block size is no longer sufficient to deliver all transactions submitted by nodes. Therefore, blockchain offers different features to smooth out transactions and follows a concept known as the Scalability trilemma [33], where one aspect is sacrificed to accommodate the other two.

#### 3.4.1. Layer 1

The term “layer 1” refers to the base level of a blockchain architecture. It’s the main structure of a blockchain network. Ethereum, BNB chain, etc. are popular examples of layer 1. The polygon (MATIC) that runs on top of Ethereum is an example of a layer 2. Within the blockchain infrastructure, scalability refers to a blockchain network’s ability to support high transactional throughput and future growth. Layer −1 solution changes the rule of the protocol directly to increase transaction capacity and speed, while accommodating more users and data. They can increase the amount of data that are continued in each block or increase the rate at which blocks are confirmed, which will then increase overall network traffic going in and out. This is directly impacted by the blockchain consensus protocols. Some consensus mechanisms are more efficient than others. Here we briefly discuss the popular consensus algorithms.

Proof-of-Work (PoW): The Proof of Work protocol is used by Bitcoin and Ethereum and many other blockchains (Ethereum will be switching to Proof-of-Stake consensus protocol by September 2022). Every time a new round of computational power competition is held, selected nodes are required to create a new block by solving a cryptographic puzzle. In a PoW puzzle, the node that solves the puzzle first has the right to create a new block, but the puzzle is hard to solve [34]. Each block is mined using the SHA-256 hashing algorithm [35] to generate a security goal (authenticity, digital signature) during mining. Due to these algorithms, Hal Finney was able to develop a concept called “reusable proofs of work” [36] that helped future generations to adapt to digital money security. The concept was based on Nick Szabo’s theory of collectibles [25] Despite their security, they are rather slow, so many new blockchain networks prefer proof-of-stake (PoS).

Proof-of-stake (PoS): The Proof-of-Stake (PoS) system was first adopted by Ppcoin (Peercoin) in 2012 [37]. In PoS, computational power does not determine which node creates a new block; rather, the stake held by the node determines how many new blocks are created by the node. Although the node still needs to solve a SHA256 puzzle [36], the key to solving this puzzle is the amount of stake (coins), as the network only validates new blocks based on staking collateral. As a result, it becomes an energy-saving consensus protocol [34]. Staking refers to pledges of your coins for transaction verification. Staking your coins locks them up, but once you’re ready to trade them, you can unlock them. The longer you stake, the more rewards you’ll receive. For PoS, a validator node reviews the block transactions to ensure they are accurate. A validator receives a reward (interest rates 2–6% a month or 10–20%) if the information is accurate; however, if the information is inaccurate, they will lose some of their stake holdings as a penalty. Table 1 compares the consensus algorithms in terms of energy consumption, hardware requirements, security, speed, and popularity. Further details can be found in the reference provided [38].

#### 3.4.2. Layer 2

“Layer 2” refers to a network that runs on top of the existing blockchain protocol to improve its scalability, efficiency, and security while maintaining its decentralization. The framework is designed to handle transactions off-chain [39] so that the blockchain itself receives less load, thus allowing faster transactions. In general, layer 2s host transactions and report only a “summary” to the main chain. This is accomplished by transferring some of the blockchain protocol transactional burden to an adjacent system architecture, which then handles the bulk of network processing and only reports back to the main blockchain to finalize its result, resulting in less traffic on the blockchain’s base layer. Several kinds of layer-2 (L2) solutions are available, including:

Sidechains: the first concept of sidechains was introduced in 2014 [40]. In addition, it is an independent blockchain network connected to another blockchain network—commonly known as a parent network or main net—by a two-way peg. Using their own consensus protocols, they can improve the privacy and security of the secondary blockchain and minimize additional trust requirements. The main concept of the main net (parent blockchain) and secondary blockchain is to facilitate a smooth asset exchange. Using two-way pegs, sidechains are linked together. Instead of “transferring” digital assets, they are simply locked on the main net while on the sidechain the equivalent amount is unlocked. Off-chain processing is used to transfer digital assets, which occurs outside of the main network. During this process, the transaction details will be relayed to smart contracts on the sidechain for verification. Only after the verification is successful will the actual funds be released on the sidechain, allowing digital assets to be moved between both blockchains.

Rollups: Ethereum co-founder Vitalik Burerin first described rollups as shadow chains in 2014. Through rollups, transactions can be aggregated off-chain inside a smart contract, reducing fees and congestion by doubling the blockchain’s transaction throughput from its current 10–15 transactions per second (tps) to more than 1000 tps. Within minutes you can move funds between different applications like fiat accounts, top-up cards like Revolut [41] to exchanges to crypto.com [42]. It is important to note that different types of rollups handle different tasks.

As of 2019, John Adler [43] was the first one to post a detailed optimistic rollups paper. As optimistic rollups are done, they ensure that transaction data submitted to the blockchain is accurate and valid, and that no spurious translation is used within rollups to fool the blockchain. For this system to work, when a transaction is directly submitted to a blockchain network, it checks if it is legit and settles a dispute, but parties stake their digital assets and lose money if they commit fraud. At present, rollups can improve the speed and smoothness of transactions through batch processing of transactions. The other concept is zero-knowledge rollups, which was proposed by Barry Whitehat in 2018 [44]. Unlike optimistic rollups, ZK-proofs allow you to mathematically prove that a statement is true without disclosing any additional information. Dispute resolution is also non-existent. ZK-proofs also validate sidechains and allow off-chain storage and processing. The major network also allows only valid transactions to be uploaded with SNARK (Succinct Non-integrative Argument of Knowledge). Furthermore, this technology makes transaction data cheaper and faster, while being less resource-intensive and just as secure as the original.

### 3.5. Transparency

Among the most important features of both NFTs and Blockchain is transparency. Blockchain is responsible for recording all digital ledger data publicly, and everyone can access it. NFTs are essential elements for digital art, and they are supposed to regulate and make all transactions transparent, wallets, and ownership relationships of digital artwork, regardless of their platform. As transparency is not always clean or decipherable, the ideal of transparency associated with NFTs is partly manifested, for example, looking back towards the beginning of that NFT, as to who owned it in the past, and seeing the movements of that NFT (wallet and remaining balance) to determine where it was taken from there.

As an example, the first CryptoKitties were priced at $3.24 each in December 2017 and 105 CryptoKitties were created with a total of $340.63 worth of ETH spent by different collectors. At present, the average price of CryptoKitties is $256.99 with an all-time average return of over 7800% [45]. In addition to transparency and sharing, this is one of the features of the blockchain. However, they do not provide any in-depth information about the user specifically. The transparency is determined by the level of access provided by the underlying blockchain framework.

#### 3.5.1. Permissionless Access

Many factors contribute to NFTs’ transparency, but the most prominent reason is the public permissionless blockchain, which serves as the basis for all cryptocurrency-related activities. The NFT system allows users to check ownership and transaction histories of all NFTs which have been issued up to the most recent transaction without needing any permission. As a result of this access, you can verify that you have bought the original work directly from the artist. Blockchains can be classified into three types: public, private, and hybrid but we will only discuss a public blockchain as it is the only one which is primarily used for NFTs; for example, the Ethereum blockchain. In fact, a few Ethereum networks do indeed have a private, e.g., permission-based, Ethereum codebase network used for business-to-business transactions [46], but all data and value are always stored on the main public Ethereum network.

In a public blockchain, also known as a permissionless blockchain, users may join any blockchain network and become nodes of the network [28], with limiting nodes’ (miners’) rights. In this blockchain, nodes create a blockchain for transactions on a network, working like a bank for a small fee. This allows users to view and validate transactions. Due to their large number of nodes and size of transactions, permissionless blockchains tend to be more secure, but they also have a longer transaction processing time. Generally, permissionless systems rely on consensus mechanisms like PoW and PoS [47] and keep data safe from security breaches, hacking attempts, etc. [48]. Permissionless blockchains are the main reason why NFTs have gained widespread acceptance and are used in a wide range of applications.

Permissionless blockchain technology allows users to trade NFTs without divulging their identities or undergoing Know-Your-Customer (KYC) [49] checks that require submission of government identification. By doing so, the user and their NFT investment can remain private and can continue to do so. This is because when a transaction is made, the user’s wallet address and the transaction are both recorded on the blockchain. In this case, if the wallet address is not linked to the user’s identity, the transaction remains anonymous. Virtual private networks (VPNs) are primarily used for this [50].

There are many assets that require significant effort and time to acquire, but NFTs do not fall into that category. NFT marketplaces (Open Sea, Rarible, etc.) are some digital platforms where NFTs can be bought and sold. On this platform, users can store and display NFTs, as well as sell them for cryptocurrencies or money. Additionally, there are several NFT marketplaces (OpenSea, etc.) in which users can mint their own NFTs, making the process of processing and rendering them faster. There is also a straightforward process for listing your NFT on most NFT marketplaces [51]. The NFT you created can be available to the public and can be sold within minutes.

#### 3.5.2. Permissioned Access

In contrast to public blockchains, this type requires a permissioned blockchain, where members are strictly controlled, information is managed, and transactions are validated. Despite being a decentralized network and doing distributed structures, this network is controlled by one organization where the central authority decides who can be a node. The focus is on scalability, and they remove the idea of full transparency, as there is no consensus and only leaders can nominate validators. There is a private NFT ownership field that is currently available, but it is not part of the Ethereum, Solana, Binance Smart Chain, or other similar platforms; rather, it is an optional private metadata field. Through these, users can conceal ownership and transactions from the public, and everything happens on a secret network [52] and works similarly to a business-to-business virtual currency exchange, such as Ripple (CRP), Hyperledger, etc. [46].

Each version of blockchain has its own strengths and weaknesses, as you can see in Table 2, with all their versions being different and each having its own strengths and weaknesses. They are just designed for different purposes, the public being decentralized at the expense of speed, for example, and the private being centralized, but powerful.

#### 3.5.3. Transaction History

Verifying transaction histories is very straightforward; depending on what blockchain you are using to buy, sell, and create NFTs, you can obtain data from a variety of sources. Every NFT purchase, sale, and cancelling the order, as well as all gas fees paid will generate a unique ID in a blockchain transaction. It all depends on the cryptocurrency used to purchase the NFT; for example, Ethereum, which is one of the most popular blockchains and comes with a tracking system called Etherscan [53]. The platform explores, searches, and analyses Ethereum blockchains. Numerous search options are available, including wallet address, transaction hash, block number, token ID, and ENS domain.

There are other blockchains as well, such as polygon—polygonscan [54] or Solana—solscan [55], that provide tracking capabilities. Ultimately, it is all about preference and what the user is looking for to accomplish their own goals, such as a system that can provide faster transactions, a budget NFT transaction, a higher volume/scaling of NFT transactions, or a more robust security transaction, etc.

### 3.6. Security

Over the past year, the NFT has grown, and a large amount of attention [56] has been paid to this area of the marketplace, which is bound to have some risks since not all NFTs are completely secure. As a first point, it is imperative to note that malicious users are prepared to exploit any assets that have value right away. The popularity of NFTs has become very widely known, so this attracts hackers to exploit NFT vulnerabilities, which can significantly undermine the security of NFTs. In 2021, for instance, hackers were able to gain unauthorized access to the accounts of several Nifty gateway NFT users [57]. In addition to stealing digital artwork worth thousands of dollars, the hackers also used their credit cards to purchase additional NFTs, which cost them thousands of dollars. A further problem with this attack is that the attacker then sold the NFTs to another buyer on a different platform and users were not able to recover their NFTs because Nifty Gateway held the private key. It is known that a user account’s public key is essential for sending and receiving crypto. Although your public key appears in the blockchain transaction, you should never share your private key. The private key contains the digital assets that you can trade or prove ownership of, as well as the authentication mechanism for those digital assets. The security of NFTs is largely determined by the security of the private keys or credentials that can be used to access those keys, for example, the NFT Marketplace. In the event that a thief or malicious attacker compromises the private key, the rightful owner may not be able to retrieve the corresponding NFT, and without your private key, you cannot reclaim or regenerate it [58]. All these events can be represented as such as people sharing the keys with those close to them or misplacing the keys, as in the case of a Polish exchange, Bitomat 2011, when an employee restarted the server, which destroyed a crypto wallet containing 17,000 bitcoins [59]. Additionally, several malicious attackers have plagiarized NFTs, making people believe they are genuine and selling them.

#### 3.6.1. Two/Multi-Factor Authentication (2FA/MFA)

This authentication ensures the identity of the digital user by requesting at least two pieces of evidence that the individual is who they claim to be. The factors must be categorically different from each other, but must be something they own, possess, or know. Using these factors when logging into an online account helps ensure that the user’s information will be protected, and even if one factor is compromised, the chance of another being compromised is low. In addition to a password, the second factor can be anything such as a security code sent to the user’s phone via SMS, voice, email, or biometric verification of your thumb or even face ID from the phone.

These standard features are available on many cryptocurrencies’ exchanges and NFT platforms as of now as they revolve around buying and selling as well as storing digital assets. In the past, authentication methods were not required and were only optional. In March 2021, Nifty Gateway user accounts were compromised [60] and indicating that the accounts which were compromised did not use two-factor authentication. Many NFT marketplaces recommend Google Authenticator [61] and Authy [62] as reliable applications for securing platform login authentication.

NFT is mainly intended to capture and claim ownership of digital assets, which are easy to access. Nonetheless, digital collectibles may also be the first sign of NFT vulnerabilities since they provide easy access to criminals. Due to the fact that cryptocurrencies lack a physical form, they’re difficult to trace when converted from cash into digital assets; also, as there is no physical form of cryptocurrency, laundering money is much easier in recent cases. Money laundering opportunities are increasing, and a BBC report said that crypto money laundering had risen to 20% in 2022 [63]. A major problem is that there is no centralized authority or regulator to oversee this. Likewise, NFTs are anonymous transactions, and while some exchanges may require KYC (know your customer) verification [49], there are many others that do not. The same applies to the NFTM. Although an investigation was conducted when an account was created, no KYC, nor AML (anti-money laundering) measures have been implemented by the NFTM.

#### 3.6.2. NFT Storage

Interplanetary file system (IPFS)—in addition, it is one form of storing NFTs on the Sidechain, which decreases the chances of it being hacked. This includes a pinata, NFT storage, etc. The IPFS [64] content identifiers (CIDs) are hashed data that is directly connected to your NFT content, making it one of the safer methods for storing your NFT, as opposed to the Hypertext transfer protocol (HTTP) links [65], which are susceptible to modification and hacking. Additionally, a CID can only ever refer to one piece of content, so users can be assured that nothing can be changed or modified without breaking the link, making it ideal for NFT storage.

A link to an HTTP address where anyone can access the content if they pay the bills, but the content of that link isn’t guaranteed to be the same as the NFT created since the server owner can easily replace it with something new. In contrast, CID addresses are accessible by anyone if at least one copy is present on the network, and if the original provider disappears, the data in that address is still accessible and smart contracts can be used to update the ULR of a token by allowing you to change its URL while keeping a record of the original version in the blockchain transaction to ensure everyone knows who, what, and when the changes were made. IPFS makes for a more secure option when it comes to NFT storage, however, they remain vulnerable to computer hacking.

#### 3.6.3. NFT Wallet

It is important for users to consider storing their NFTs in a highly secure location since they don’t just cost money, but also hold sentimental value. This is where crypto [66] wallets come in. These wallets are either physical devices or computer programs that allow users to store and transfer digital assets. The wallets are based on the blockchain protocols that NFTs are built on, with currency support for users to buy NFTs in, such as Ether. Rather than storing cryptos and NFTs, crypto wallets [66] hold a pair of public keys (user addresses on the blockchain) and private keys (proofs of ownership of the funds) that you can use to manage and transfer your assets. Due to the fact that they exist on the blockchain, rather than in your wallet, they offer an ease of interaction with cryptocurrency and NFTs in the blockchain.

In a crypto wallet, a series of words known as a seed phrase or recovery phrase is automatically generated at the beginning of the wallet creation process [67]. This seed word is then used as a base for creating a private key. This information should never be shared with anyone, otherwise they would have access to one’s digital assets. Like a password manager, it is a way to access digital assets and recover them if they are deleted. In the wallet category, users will find two kinds of wallet: “Hot wallet” and “Cold wallet”.

#### 3.6.4. Hot Wallet

Also known as software wallets, these applications can be downloaded and installed on your computer, device, or any other electronic device. These include MetaMask, Wallet Connect, etc. A software wallet is generally more convenient and easier to use than a hardware wallet since it stores private keys online and is always connected to the internet. The use of a software wallet facilitates transactions on many marketplaces, making it easier to trade. However, since they are continually connected to the internet, they are more prone to scams, and malware attacks. Even with high-profit security measures, online wallets that store users’ security keys and codes remain vulnerable to hackers. A small mistake can wipe away all the assets forever. The best option in such situations is insurance [68].

#### 3.6.5. Cold Wallet

This type of wallet is also known as a hardware wallet and is essentially a USB stick that stores crypto keys and NFTs. Among these are Trezor and Ledger. With a cold wallet, it is necessary to physically plug into your computer to access your wallet, as all digital assets are stored within the device rather than online. The reason that they are often considered to be “cold storage” and far more secure than hot wallets (software wallets) is that they can be completely isolated from the network. The other thing is that even if these devices are connected to the Internet, the items that are stored will be difficult (but not impossible) to compromise. This is because transactions are signed with the private key on the device and then the signature is broadcasted to the network. As a result, the private key will never leave the USB device. Due to this, malware, keylogger hackers, etc. won’t be able to receive the information required to falsify signatures (43). As far as wallets go, cold storage wallets are regarded as the most secure and impenetrable.

## 4. Top Five Blockchains for NFTs

NFTs are not all created equal and choosing a blockchain for an NFT can be extremely difficult. Unlike other blockchains, NFTs are non-fungible, which means prior assets do not appear on level blockchains, so there is also a difference in importance of mining based on the block times of each blockchain (mine a new block), which are all determined by the person who mines the NFTs, as shown in Table 3. There is no definitive answer to the question of which blockchain is best for NFTs, but it can give you a general idea of which blockchain you would prefer to invest in when you begin.

### 4.1. Ethereum

Ethereum is the largest and most popular blockchain for NFT projects [77]. NFT marketplaces such as OpenSea, Rarible, SuperRare, and many others are featured on the site. The system is highly decentralized, providing finance (DeFi) and legal services without the need for intermediaries. It also comes with technical documentation that helps developers build smart contracts. Several NFTs are based on this blockchain, including the Ethereum Request for Comment ERC-721 token [78]. It is the foundation of most existing NFTs such as Bored Ape Yacht Club (BAYC), HashMasks, and many more, and will be used in many future ones [79].

However, Ethereum’s popularity has come with its own set of drawbacks, such as its high transaction fees and energy usage. The main reason is Proof-of-Work (PoW) [39], which requires computers to compete with one another to solve complex puzzles in order to add blocks and transactions to the blockchain. It has been estimated that Ethereum will switch to Proof-of-Stake [80] around August of 2022, allowing the network to scale better and transaction speeds to increase; reducing their costs at the same time will be known as ETH2/Serenity, and is part of Ethereum 2.0 [81].

A short time later, on 6 September 2022, Ethereum’s two-stage merger, codenamed Bellatrix, began, which has been completed, as well as the post-merge on 16 September 2022. Despite the success of the switch to proof of stake networking, ETH prices plummeted shortly thereafter by 8% to $1485; however, Glassnode predicts this drop was coming [81]. The switch has, however, reduced the network’s energy consumption by over 99%, making it more of an energy-efficient blockchain that is more environmentally friendly despite no change in Ethereum speed or scalability.

TPS performance: With sharding and Proof-of-Stake, it can process anywhere between 20,000 and 100,000 transactions. Moreover, Vitalik Butherin, the co-founder of Ethereum, has stated that Ethereum 2.0 will allow the primary smart contract platform to scale to 100,000 TPS according to the roadmap he has outlined [82].

The Ethereum price fluctuates frequently and may increase or decrease within a short period of time. In any case, some analyses have been conducted that provides some of the forecast values for the future, and Anton Kharitonov of TU predicts Ethereum prices will range between $960.640 (the minimum price) and $2508.528 (the maximum price) by the end of the year, bringing the price to $1642 with its current volume at $17,781,331,143. Ethereum’s price is expected to increase up to $4781 (maximum) by 2025, and to $36,815 (maximum) by 2030 [83].

### 4.2. Solana

After launching in 2019, Solana (Yakovenko n.d.) is one of the biggest candidates to potentially outsell Ethereum with NFTs. There are over 400 projects in NFTs, DeFi, and general Web 3.0 applications. Aside from its high performance [84], it is one of the fastest programmable blockchains in the crypto space because it can handle high volumes of transactions per second (TPS). Due to its low carbon content, which is one of the biggest concerns addressed in blockchain technologies, it provides a robust infrastructure to mint and store NFTs in an eco-friendly and cost-effective manner.

A unique feature of Solana is that it combines Proof-of-history (PoH) [85] with Proof-of-Stake (PoS) [37], which gives a hybrid consensus algorithm [80]. During PoH, the Blockchain is extremely fast, the transactions are ordered correctly, and the leader (Validator) finds the transactions correctly. It also reduces the messaging overhead in a Byzantine Fault [86] replicating state machine while maintaining good security. In addition to its stability, the Solana ecosystem is one of the fastest-growing NFT platforms, like Solanary and Magic Eden.

Due to its lack of widespread use, Solana has fewer NFT markets and fewer people trading on the Blockchain. Despite it being relatively new, Solana still shows great promise and with its high performance and low fees, it deserves attention as a mining and trading ecosystem for NFTs.

TPS performance: on average, Solana network currency processes data at a speed of 3000 transactions per second. In spite of this, it can provide up to 65,000 transactions per second due to a feature called byzantine fault tolerance that reduces time and increases efficiency. In the future, it is expected to be capable of reaching 710,000 transactions per second [87].

A price decline of 8.88% has been recorded in the past 7 days for Solana. According to analyses, the price will be at least $33.43 and as high as $36.63 in 2022. With a trading volume of $1,406,822,560 as the year carries on, analysts have predicted a minimum price of $65 and a maximum price of $85, by 2030, and further down the line, the minimum price will be $160, and the maximum price will be $200 [88].

### 4.3. Polygon

The Polygon project is based on Ethereum and offers scalable, secure, and instance-based transactions. As well as being known as layer 2 [89] solutions for Ethereum, they extend Ethereum and inherit its security guarantees, and are also known as sidechains. Polygon is a multi-chain ecosystem that allows new products to connect to Ethereum and every valid Ethereum address is also a valid Polygon address, in addition to Ethereum compatibility, scalability, security, and developer experience.

In various markets, like OpenSea and others, Polygon has been used to sell NFT projects. Due to its scalability and compatibility with Ethereum virtual machines (EVM) (Ethereum 2022), this blockchain can build scalable solutions that are also compatible with faster transactions. As Polygon is a dynamic platform, there are no upfront costs when you create an NFT, but there is a fee when you sell it. Since many people are unfamiliar with Polygon, there are a limited number of people willing to buy NFTs built with Polygon, as well as their reputation for being challenging for beginners.

Despite Polygon being a partnership, it still uses the ERC-20 tokens [90], which are fungible tokens based on Ethereum as its main chain. According to current knowledge, even Ethereum 2.0 [10] will not struggle to maintain its top speed and price due to the fact that Ethereum is still not able to reach Polygon’s speeds. Furthermore, Ethereum and Polygon should be seen more as synergists than competitors since Polygon can connect Ethereum interoperable blockchains and boost transaction speeds and reduce transaction costs. In addition, Polygon is still introducing new scaling solutions, such as Zero-Knowledge and Optimistic Rollups, with the goal of creating the Internet of Blockchain.

TPS performance: Polygon currently offers the fastest transaction speed, which can reach 65,000 transactions per second. It is layer 2 of Ethereum that provides a hybrid environment. The other thing to keep in mind is that even with the Ethereum 2.0 merge, it will not be able to beat Polygon’s TPS.

The current trading volume for Polygon is $242,728,655. Trading Beasts estimates MATIC will reach $1.17 by year’s end, but there is a possibility that the price could drop. However, by 2024 there is a possibility that MATIC prices will rise again, potentially reaching a $2.2 maximum with an average price of $1.8 and in 2030 there is a possibility that MATIC prices will reach a maximum of $19.14 (maximum) and that prices will average around $16.26 on average [91].

### 4.4. Binance Smart Chain

In 2017, Binance [92], the world’s largest crypto exchange platform, was born. Three years later in 2020, the Binance smart chain (BSC) was launched. The Binance Chain runs alongside BSC, adding Ethereum smart contract support for transactions, making it a popular Ethereum alternative. NFTs can now be created using their own token standards, such as BEP-721 and BEP-1155.

This blockchain (BSC) is now known as BNB (build and build) chain network and it supports Ethereum DApps and tools, which makes it easier for developers to import projects from Ethereum blockchain with ease. As BNB chains are superfast, they can accommodate high levels of transaction rate with low fees in most of their NFTs projects.

Despite this, BSC is largely centralized, and centralized systems are more prone to system failure, threats, regulations, and even hacking. However, the consensus model [79] known as proof of staked authority (PoSA) [93] has a short block time and a low fee, allowing for a competitive advantage for the BNS and the BSC NFT market started picking up steam by the end of 2021. Soon, they plan to bring all different types of projects together under one umbrella, such as Metaverse, DeFi, Web3, NFT, and more. Meta-Fi [94] is a combination of the meta ecosystem, and DeFi and are also mainly possible because of multi-chain functionality and bridges.

As they are a fast and affordable blockchain for one’s project, BNB NFTs is a good option to exchange NFT earnings without breaking into other apps.

TPS performance: There are only a few cents charged per transaction in Binance smart chains, and their network speeds reach around 300 transactions per second. As of now, Binance has transitioned to a new blockchain network known as BNB chain, which is currently capable of supporting 100 million addresses and 10,000 transactions per second [95].

The current trading volume of BNS is $1,029,723,473. The coin is currently trading at $273.82, and the prediction is that in the long run, it can be a profitable investment. In 2025, its price is expected to increase by a massive amount, reaching $2394 at its maximum and $840 at its minimum. It has been predicted that coin prices could even increase to $15,622 at the highest and $10,949 at the lowest by 2030, based on input from multiple sources including Trading Beasts, Wallet Investor, Bitcoin Price Forecast, etc. (Education 2022).

### 4.5. Cardano

Founded by Ethereum co-founder Charles Hoskinson, Cardano is a distrusted proof-of-stake blockchain. With its 2017 release, it has become one of the top five cryptocurrencies on the market. It was primarily created by experts in the blockchain industry without whitepaper and was peer-reviewed. With its platform, transactions are settled in its native currency and scalability is prioritized. Additionally, cardano is an eco-friendly blockchain.

Its slow upgrades have been criticized recently, and the value growth in NFT trading hasn’t been spectacular. In accordance with an analysis done by Adilin Beatrice (Beatrice 2021), they start at $2 in 2022 and rise to $4.1 by year’s end so it is unclear if Cardano will rise above or fall short because of their poor value, which could lead to them falling short.

It did, however, gain some traction in November when it hit 200,000 transactions per day, and they are currently working on optimizing the growth of the network and scaling to handle more TPS [96]. Aside from this, they are also developing a layer 2 scaling protocol called “Hydra”, which will be able to scale the wider Cardano blockchain by reducing latency, transactions per second, and transaction costs [97].

Similar to Binance’s blockchain, Cardano uses Ethereum Virtual Machine to support Ethereum-based smart contracts and DApps. This is a great starting point for newcomers to start minting NFTs as it has low gas charges, and fast transactions, as well as being the most accessible blockchain for those interested in minting NFTs. This is due to the fact that the API for working on the blockchain is provided as a free community service, allowing them to communicate directly with the blockchain nodes or client networks.

TPS performance: The Cardano scaling solution hydra will be a game-changing upgrade because they have previously said it will be able to handle 1 million transactions per second (TPS), but nothing is certain yet. However, Cardano has increased its block size by 12.5% to allow more transactions per block, resulting in a higher transaction rate [98].

As of right now, Cardon has a trading volume of $937,024,032. Based on price predictions for 2022, the minimum cost of cardano will be $0.419, while the maximum will be $0.44996. According to the previous estimates, prices might drop to $1.27 as of 2025, but there is a possibility that it can reach $1.62. According to the various crypto exerts, the coin price is estimated to average around $8.41 by 2030 with a maximum peak of $9.41 [99].

## 5. NFT Marketplace

NFT marketplaces (NFTM, or NFT trading platforms) are platforms to buy, sell, and trade NFT assets, and this includes OpenSea [100], which is the largest NFT platform with a trading volume over $5 billion between Ethereum and polygons [101]. Other NFTMs, such as Rarible and SuperRare, are also available for buying and selling NFTs. According to Table 4, each of these marketplaces has its own pros and cons, as well as its own market value for money, which makes them the best trading platform.

People are increasingly interested in digital assets these days. From a purely economic standpoint, the reason people are so interested in the NFT ecosystem and market is that they outperform cryptocurrency in sales volume and conversation. As the wheels of the NFT movement have turned and grown in this recent month, the number of NFT buyers has skyrocketed. In late 2017, the popularity of collectible CryptoKittes led to the first significant growth of NFTs, with the total weekly sales increasing from 100 to an average of 15,000 to 50,000. This occurred after around 22 November 2017, when a rare cat was sold for an extremely high price, and since CryptoKitties have so many attributes, it is difficult to put an actual value on them, but due to their rarity and Gen 0 status, they were worth over 200 Ether, and many people started looking for their own rare cats. Within the next few years, the sales went from 30,000 to 80,000 per week [102]. According to Manas Sen Gupta [103], the most expensive NFT sold on the Nifty gateway in January 2022 was nearly $91.8 million.

Considering the high level of sales, it should come as no surprise that so many people want to participate. The NFT marketplace is the gateway to buying and selling these digital assets.

### 5.1. NFT Ecosystem Development Structure (Creator, Buyer, and Seller)

NFTs are commonly used to sell digital collectibles (images, files, audio, video, etc.) and can be sorted, shared, and traded through emerging NFT marketplaces. To accomplish this, there needs to be a transparent, immutable ledger (a more secure platform for storing, visualizing, and managing both coins and NFTs) that logs who owns what and where the files are located. The blockchain secures all of these, including transactions, cryptographically. As part of the blockchain, there is a digital proof of ownership, and this ownership is represented as tokens for all NFTs/digital collectibles. In addition, it is because of the blockchain that we can buy, sell, and trade these digital goods without the fear of double spending [104] or tampering with past transactions. Figure 4 illustrates the overall economy that has developed around NFTs, specifically the participants and the components that they interact with.

#### Components Interaction

The ecosystem of NFTs is divided into three parts: content creators, sellers, and buyers. As you may be aware, content creators create digital content from raw data and upload it to hosting services. Host services are an external database outside of the blockchain that makes raw data (digital content) available to the public. Some content creators are not technically skilled enough to convert their work into an NFT and sell it on the blockchain. This is where marketplaces permit sellers to mint NFTs for sale, but creators can also sell them if they have the skills. Upon listing the digital assets on a marketplace (a decentralized app platform) [105], buyers can make offers, trade-in, bids, or buy the artwork for a listed price. Using a previous diagram from the research report NFT-eco system [106], this diagram was created.

A decentralized app platform is an essential component of the NFT ecosystem, through which NFT assets are traded. Users can send transactions through the web app to a smart contract that implements the NFTM protocol and blockchain, as well as a token contract on their behalf, allowing them to perform a variety of tasks, such as mining and trading, and each of these tasks is referred to as an Event. In NFTM protocols, these events can be stored off-chain, on-chain, or in a hybrid chain [106]. If stored **On-chain,** all events related to the NFTM are live on the blockchain, which means that tokens and images are validated by an attribute-based signature (ABS) [107] and directly storing the address and log there, the costs associated with operating the NFTM are increased due to the on-chain model. SuperRare, CryptoPunks, and other marketplaces use these models. If stored **Off-chain,** all events are recorded in a centralized database (cloud) and are encrypted; that encrypted data is then stored in an off-chain database (apps) rather than directly on the blockchain, making it less expensive for the user to operate in NFTM. These models are utilized by marketplaces such as NFTrade, Nifty, among others. If stored on a **Hybrid-chain,** this hybrid model combines both On-chain and Off-chain events, ensuring data transparency and integrity while connecting On-chain and Off-chain with cryptographic checks, providing the level of decentralization and security that users expect. These models are used by marketplaces like OpenSea and Rarible.

## 6. Challenges Faced by NFT Ecosystems

### 6.1. Security Issues

Several serious security incidents have occurred because of the influence of the Non-fungible Token (NFT). Since they are one of the most well-known digital assets right now, they are bid on and sold for millions of dollars. According to OpenSea, which is one of the largest marketplaces, 2.4 million NFTs [108] were sold in January 2022. As a result, malicious users can also be an issue with this kind of attention. In March 2021 [109], the number of suspicious-looking domain registrations with names of NFT stores such as ‘Rarible’ and ‘OpenSea’, among others, increased nearly 300%. 

NFT marketplaces require cryptocurrency wallets to participate. The challenge here for NFT holders is that attackers can find their way into users’ crypto wallets through the marketplace accounts they have created. The NFTM platform is also more like a centralized platform, similar to Open Sea and Nifty Gateway, where all assets on the platform are controlled by them. If the platform/Marketplace is compromised, hackers could take large amounts of NFTs just as they did with Nifty Gateway [57]. The following subsections describe the popular security threats impacting NFT ecosystems:

#### 6.1.1. Phishing Attack

The easiest tasks for malignant actors are often those that require low effort and are often automated, such as phishing and social engineering. Moreover, these types of attacks a wide range of users because most NFTs or high-value operators are well-known, which makes it easier for malicious actors to study them.

The purpose of phishing attacks [110] is to trick victims into opening emails, instant messages, downloading links, etc., leading to malware installations (viruses, worms, spyware, etc.). An early February 2020 phishing attack resulted in the theft of $2.9 million worth of NFT tokens from 17 OpenSea users [111]. Co-Founder and CEO of OpenSea Devin Finzer [112], explained how the attack worked; first, the target signed a partial contract, leaving a large portion blank. After the NFTs were already signed, the attacker completed the contract by calling it their contract, which transferred ownership of the NFTs without payment.

Possible Solution: There are many different kinds of Phishing attacks, and the solutions for these kinds of attacks vary. To mitigate this risk, never click on links that seem random or suspicious. Use an anti-phishing toolbar on your electronic devices to prevent malicious attacks and install a firewall to protect your device from untrusted networks. It is a smart idea to regularly check one’s account for fraudulent transactions and perhaps change passwords as well. On 18 February, OpenSea was preparing to implement a smart contract upgrade that asked all users to migrate NFT from the Ethereum blockchain to a new smart contract. As the upgrade was happening, hackers launched a sophisticated phishing campaign, which allowed them to steal NFT from the storage platform. During the investigation, it was discovered that the attack originated outside of OpenSea website. This was done by sending fake emails that clearly stated that users should be educated and that marketplaces would never contact users via email besides registering with them. Another possible solution would be to use hardware wallets since software wallets and other custodial solutions are too vulnerable in their design and operation externally [113].

#### 6.1.2. Scams or Rug Pulls

In rug pulls, a team pumps a project’s token, then disappears with the funds, leaving investors with worthless assets. Essentially, this issue comes in two forms: hard pulls, in which developers walk away or add malicious code to tokens from the start, which affects the token value (liquidity pulls), and soft pulls, in which creators sell large amounts of tokens or sell them in increments, lowering the value and making them worthless. Due to the free market, developers are not illegally selling their tokens, which makes them harder to identify.

This ice-cream-themed NFTs collection had built a sizable community by marketing itself as “cool, delectable, and unique” and priced their NFTs at 0.004ETH per mint. After the entire collection of that project was sold out, the founder [114] withdrew 335 ETH (just over one million dollars) and disappeared with the funds for sales being transferred to various wallets. The investors/people who bought the collection were left with nothing but their detail art after the fund was transferred. It was reported that the two defendants were arrested on March 24 and have been charged with fraud to commit money laundering. The case continues to be reviewed [115].

In Solanart Marketplace, the first and largest fully fledged NFT marketplace mined using Solana blockchain, another rug pull occurred. There were over 2222 ape-themed NFTs included in the collection titled Big Daddy Ape Club (BDAC), but things went sideways here [116]. It is reported that the project collected 9136 SOL [117] which is just over a million dollars and that when the time came for mining the NFT collection, BDAC locked and decommissioned their social media (community), and investors did not receive the NFTs they had paid for. A sad part of this case is that Civic, which is a decentralized identity verification company, had verified the project once and this was the third rug pull [111].

Possible solution: The first step to avoiding scams or rug pulls is to identify the RED FLAGs of the project prior to investing. Look at the community and personal information of the team and check their social media for updates to determine the status of their development. Due to the fact that they can have fake Twitter or Discord followers or accounts, it is a smart idea to do a quick check. For example, if an account has 120 k followers and only 50 likes, this is a red flag [118]. Low trading volume, liquidity, and the community of the product can indicate a rug pull scheme. The roadmap of a long-term, scalable project can help further understand their agenda. Use NFT Explorer to analyze the purchases and merits associated with the wallet ID. Rug pulling is defined as a series of incidents like those in the situations of the big Daddy Ape Club, which stole $1.3 million after the creator vanished, along with all the accounts being shut down, or Blockverse, which sold out 500ETH (more than a million dollars) [116] and was shut down after a few days. In light of all these incidents, people are now being asked to first look for red flags, which indicate that the project being built isn’t genuine. In the world of NFTs, there is a saying that if a roadmap (project) sounds too idealistic to be true, then it probably is; therefore, one should invest as much as one can afford.

### 6.2. Transparency Issues

As an asset-ownership record, NFTs that can be stored on the blockchain by using smart contracts as blockchain technology are extremely transparent, allowing one to view the total amount of coins and volume of transactions. The NFT project is divided into 2 parts—the smart contract and the metadata for the artwork itself. A smart contract exists on a blockchain (Ethereum, Solana, etc.) and is governed by rules and regulations that facilitate transactions and provide a description of the artwork’s content. 

#### 6.2.1. Lack of Transparency

Conversely, digital artwork is stored outside blockchain, or “off-chain”. With off-chain servers, images are centralized within the Interplanetary file system (IPFS) or a company. One problem is that if the server goes down, the images will be lost, thereby losing the NFT collection. Aside from the content of the artwork, each transaction within the marketplace includes information regarding each transaction: the address of the seller, the price, the time of transfer, and the price at which it was sold. Due to the fact that all these records and transactions are stored off-chain, it is impossible to verify any trades or ownership history. A malicious NFTM could take advantage of this situation by forging spurious sales records, tampering with the records, censoring, etc. if the NFTM database were to collapse. Some of the marketplaces that maintain off-chain records include OpenSea, Nifty, NFTrade, and Fauna [106]. In essence, when NFTs are listed in a marketplace, the marketplace will be in control of the NFT by transferring it to an escrow wallet. With this wallet, the marketplace can temporarily hold assets for trading purposes, but trades aren’t recorded on the blockchain because they are off-chain. After that, owners can decide when to withdraw NFTs from the marketplace and NFTM will transfer the tokens back to their accounts.

Possible Solution: In some cases, these issues are also about trust. Do you believe that NFTs are legitimate markets, or just the ones you’re buying from, and did the organization behind them ensure reliable access? These are the questions you should ask before purchasing an NFT project. Two main blockchain-related differences exist among these marketplaces: 1. The public network is extremely secure (immutability because the hash function serves as a security protocol), public, anonymous, and fully decentralized. Finally, there are no regulations, and it is completely transparent, which means that corruption and discrepancies are impossible. 2. In private blockchains, all information is private. Users are protected from unauthorized access, the service is stable, and transaction fees are low. No illegal activities are permitted, and regulations are clearly outlined. In both cases, each has its own pros and cons, and it is up to the user to decide which one will be most suitable [119].

#### 6.2.2. Royal Distribution and Marketplace Fee Evasion

Peer-to-peer trustless transactions have been possible due to blockchain technology. With this technology, every transaction is recorded on the public network and cannot be changed because of its immutable digital ledgers. As an added bonus, creators can also pre-programme their royalty condition into their work. EIP-2981 [120] an NFT royalty standard that supports royalty [121] payments, enables them to take control of where their work goes and how much they are paid for it. Even though royalty payments have been established, there are still some problems.

It Is currently difficult to integrate different markets within the NFTM since they are not designed to be compatible with each other. Despite the fact that smart contracts enable peer-to-peer transactions, they are unable to communicate across different blockchains, so royalties set on one NFTM platform are not visible to others. By exploiting this lack of coordination, malicious sellers may be able to evade royalties by trading NFTs on platforms where the royalty is not set, and the creator cannot prevent buyers from listing the NFTs on other marketplaces, resulting in the sale of every NFT and missing out on these sales royalties [122].

Fee evasion is another issue, as the ERC-721 token contract does not enforce the royalty or marketplace fees, so even though NFT royalty fees are incorporated into the NFT contract themselves, they are completely voluntary since there is no way to distinguish between a transfer and a sale, and only marketplace contracts/off-chain infrastructure pay creator royalties. Therefore, malicious sellers have the option of transferring NFTs directly to the buyer to avoid this payment or settling the payment off-platform. They can also be levied (Tax), making the API [123] transfer contract more expensive.

Possible Solution: A solution for royal distribution and marketplace fee evasion is to use the universal token standard. In doing so, creators are able to keep their royalties and have them accepted across all marketplaces. This ensures they receive credit for the work they have put into their creations. New generations of smart contracts designed to facilitate new types of transactions, such as subscriptions, can be used to share future sales of NFTs with creators [124]. Co-founder of Solana, Anatoly Yakovenko, suggested creators should be able to add instructions on their NFT contract to “freeze assets” to punish royalty cheats [125].

### 6.3. Scalability Issues

With the growth of NFTMs in 2021, the level of interest and financial gains in NFTMs has been immense, and as most people are aware, NFT collectibles are minted and traded using the Ethereum blockchain network, with the most popular marketplace being OpenSea and SuperRare. An NFT, however, despite its price range from $10–$10 million, is not an immutable asset, but rather an immutable link that points to a location where the assets are held on Wallet or IPFS.

#### 6.3.1. NFT Storage

During research into the structure and storage being offered on the platform, it was discovered that there were two types of data being offered on the platform: on-chain and off-chain, a result of scaling limitations and the cost of supporting large amounts of on-chain content data because the more often the network was used, the more expensive it became. As the price of each transaction increases, so does the cost of storage, which is why many marketplaces use off-chain storage, such as SuperRare, which collects and trades digital art. It is indicated that the assets’ metadata that is associated with NFTs on SuperRare has been stored on Interplanetary Files (IPFS) indicating that these assets were created authentically by creators (artists) [126]. This form of storage offers some scalability and cost savings, but it also presents some issues which you have already seen in the lack of transparency section.

Possible solution: Use smart contacts to store files in a network of storage via peer-to-peer sharing. For example, multiple copies of a file can be kept. In the event that one storage location is unavailable, the data is still available elsewhere. Aside from this technology, there is also Distributed Hash Table (DHT), which stores data in a storage location purchased by the consumer. In addition, there are other options available for storing NFT, including Software wallets, which are encrypted with a password and done by a browser, Interplanetary File Systems, which act as an off-chain storage solution for NFT, reducing the risk of a hack, as well as the Hard wallet, which is the most secure method of keeping NFTs and your data offline with a password that makes them unbreakable [127].

#### 6.3.2. Gas Fees

As far as the NFT marketplace is concerned, gas costs are currently the biggest issue, especially when minting NFTs on a large scale that requires storing their metadata. Due to the computational power and storage requirements of blockchain networks, even a straightforward transaction can be very expensive. A huge gas fee is currently required for Ethereum NFTs users to purchase NFTs on the Ethereum network, which is a difficult experience for them. The gas cost for all NFT transactions skyrocketed to $3300 during the recent Bored Ape NFT sales [128]. Even for $20 NFTs, there was still a $3300 gas fee for all NFT transactions. It was also one of the major problems with blockchain for the past year, where the gas costs were higher than NFT prices, causing people to miss out on purchasing.

Another purchase cost $44,000 in gas fees for a plot of virtual land known as OtherDeeds which will soon launch a metaverse game called Otherside and was purchased by Yuga Labs (the company that created the Board Apes Yacht collection) [129]. As the gas price spikes, the growth of new NFT marketplaces is also slowing down. Due to the fact that high gas prices are hurting the NFT marketplace, many platforms are starting to work on layer 2 solutions, and some are considering leaving Ethereum for another chain [130].

A possible solution would be to use layer 2, as the polygon is the largest on the Ethereum network right now with its sidechain. Immutable X is another marketplace that uses Immutable X as layer 2 and offers zero gas fees for its on-chain NFTs. The other way to reduce the cost is by choosing the right timing, such as the peak period of the Ethereum network where gas costs 40 GWEI and the low point 8.1 GWEI (on the same day). If time isn’t a concern for the user, then choosing a slower transaction may actually help you to save on gas fees; however, it’s critical not to make it too slow [131].

## 7. Conclusions and Future Direction

This article reviews the pillars of NFT and NFT marketplaces and analyses them in terms of their security, scalability, and transparency. An evaluation was conducted on the top 5 blockchains for NFT, as well as the top NFT marketplaces. The challenges facing the NFT marketplace in terms of scalability, transparency, and security are also reviewed. Currently, the NFT is being adopted into a growing number of fields and has created several opportunities for the industry. For this reason, many blockchains are now adopting and developing layer 2 protocols, which will speed up transactions and reduce gas charges. They are also working to have two or more authentication factors and verification methods, in order to mitigate the majority of security threats. This would increase the size of the NFT market and allow NFT mechanisms to be applied to various new domains. Some of the future NFT applications that would impact the mass populations are summarized below:

### 7.1. Ticketing

There are many forms of NFT ticketing, from physical paper to smart cards/wristbands with electronic codes, which can provide access credentials to any event such as sports or culture. Ticketing systems allow customers to purchase tickets directly from the event organizer or from authorized sellers on the primary market. By creating tickets with NFTs, consumers are protected from malicious invaders because they have more control over the resale market [132] than they do with normal markets, which employ bots to drive up prices automatically to earn profit by reselling them at the highest possible markups [133]. In addition to secure ticker storage, tickers can now be viewed as digital collectibles as well. Kings of Leon, for instance, sold NFTs entitling buyers to lifetime front-row seats at their tours, which is an innovative use of NFT ticketing [134]. The ticketing application improves validation, transparency, automation, and cost efficiency by representing uniqueness and providing more information [135].

### 7.2. Gaming Industry

Throughout the history of the video game industry, players have always had this need for digital properties whether it is skins for customizing the game or in-game items (all of which are extremely rare) and the ability to trade them, but there is no alternative to trading them in-game, so most people just sell their whole account, which is linked to their email address, [136]. In fact, many video game companies have already taught gamers how to accept the concept of digital ownership. The number of gamers in the world is estimated to surpass 3.24 billion by 2021 [137]. Considering this many players and the coming of NFT gaming platforms such as crypto games like Axis Infinity [138] as well as other NFT-based games that are conquering gamers’ hearts; NFT technology has endless possibilities for gaming applications. As an example, “NFT” represents all digital assets such as in-game items, skins, maps, card collections, and more, and all of them are unique. The tokens can then be exchanged for cryptocurrencies, or real money, depending on how much people are willing to pay. A study shows that 69 percent of Fortnite players spend an average of $85 each month [139] and another survey shows that the average monthly game spending is about $86, and the average lifetime spend is $60,048 [140]. Hence, there are opportunities to trade resources, which not only increase the value of the game, but also provides a kind of investment opportunity.

### 7.3. Metaverse

A virtual reality where people live, work, shop, and interact remotely. The combination of these technologies indicates what’s next for the internet. The Metaverse was originally developed for online multiplayer games, but now both gamers and non-gamers can benefit from it. The Metaverse of Decentraland [141] is a 3D virtual platform where you can buy and sell land. Decentraland is governed by only two types of tokens. Land is a decentralized virtual space in which parcel ownership is defined by NFT. MANA is a cryptocurrency that can be used to purchase LAND and digital goods. The project was established in 2014 and released in 2017 by Ari Meilich and Esteban Ordano [142]. The architecture consists of consensus layers for maintaining the ledger, where each land has a unique coordinate. Additionally, LAND tokens are available in the builder and marketplace. With these, users can exchange names, lands, and items for their avatars with MANA. In [143], it was reported that a virtual real estate plot was sold for $2.4 million in cryptocurrency. Sandbox is another metaverse platform. The platform provides users with the ability to create, own, and market games using SAND tokens that run on the Ethereum Blockchain. It is even possible for them to create NFTs that can be published and featured in games made by artists. Axie Infinity [138] is one of the most popular and influential metaverses of the year. It consists of a video-game-based metaverse in which an animal-like avatar fights and is rewarded with Smooth Love Potion icons (SLP) that can be used in Axie Infinity, the metaverse, or NFTs. The number of platforms is growing [144].

In addition to these, there are many more areas where NFTs have the potential to benefit, but I believe these are some of the closest to being achieved and are those being worked on by many now.

## Figures and Tables

**Figure 1 sensors-22-08833-f001:**
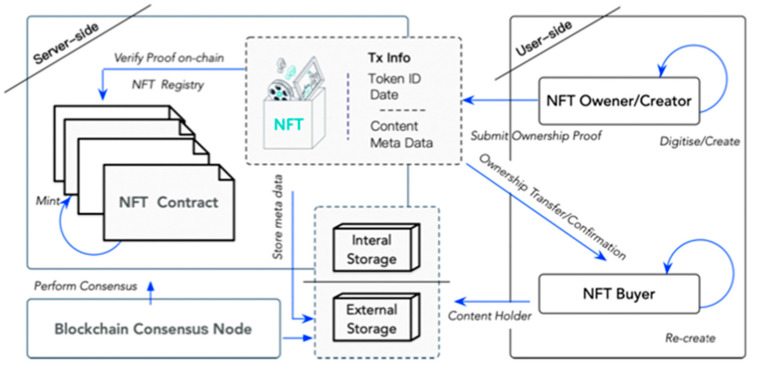
A standardized workflow for NFTs.

**Figure 2 sensors-22-08833-f002:**
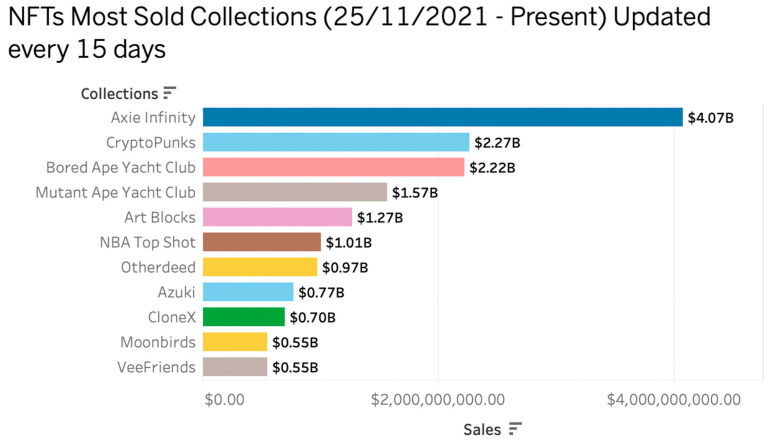
Visualization of Top 10 NFT Collections based on sales volume.

**Figure 3 sensors-22-08833-f003:**
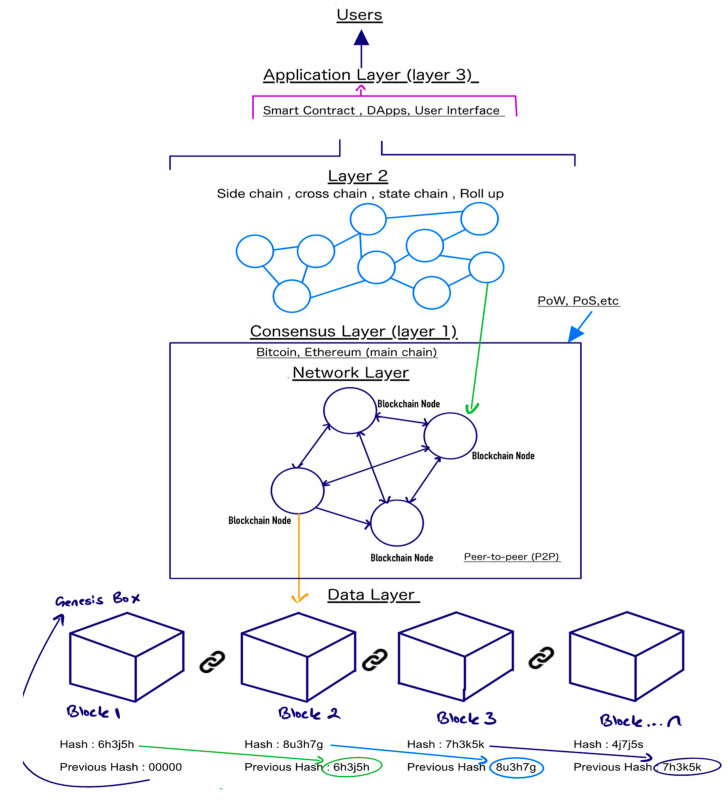
Blockchain Architecture.

**Figure 4 sensors-22-08833-f004:**
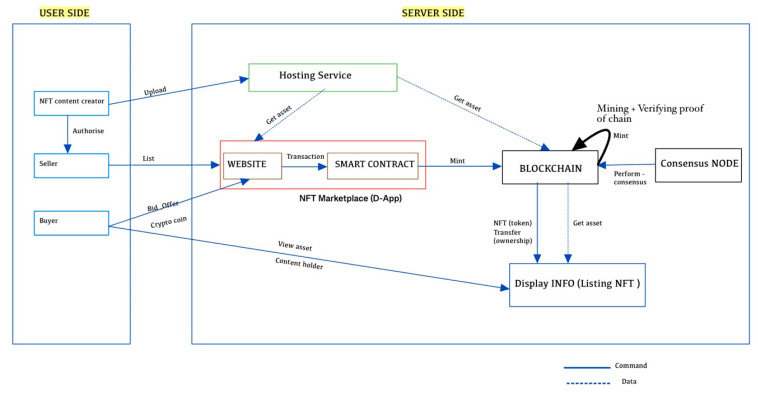
Schematic diagram of NFT ecosystems.

**Table 1 sensors-22-08833-t001:** PoW (proof-of-work) and PoS (proof-of-stake) comparison.

	PoW (Proof-of-Work)	PoS (Proof-of-Stake)
Energy consumption	Energy inefficient	Energy efficient
Hardware required	Mining equipment required	No equipment necessary/depends on stake in the network
Block generation speed	Medium	High
Security	HIGH (Malicious attacks require 51% of computer power)	Medium (To hack a blockchain network, an attacker needs to hold 51% of the stake)
Transaction confirmation speed	Low	High
Applications used in	Ethereum (But soon its switch to PoS), Bitcoin, ZCash, Dogecoin etc.	Cardano, Avalanche, Tezos, Polkadot etc.

**Table 2 sensors-22-08833-t002:** Type of blockchain access for NFTs.

	Public	Private
Permissionless	Yes	N/A
Read	Anyone	Individuals who are permitted and authorized
Write	Anyone	Individuals who are permitted and authorized
Ownership	Anyone	It is controlled by one authority or one individual
Who can Participates	Anyone	Parties with authority only
Transaction speed	Low, Bitcoin, for instance, can handle 7 tps and Ethereum can handle 20 tps.	High, Processes thousands of transactions per second, visa, for example, can process 24,000 transaction per seconds
Transaction cost	High, Due to the vast number of nodes on the platform, performance is limited by the time it takes, resulting in a higher price.	Low, because no matter how many people request transactions, the fee for private transactions stays the same regardless of the number of requests.

**Table 3 sensors-22-08833-t003:** Top five blockchains for NFTs based on NFT transaction volume.

Blockchain	Ethereum	Solana	Polygon	Binance Smart Chain	Cardano
Currency	Ether	SOL	MATIC	BNB	ADA
Consensus	Proof-of-Stake (PoS)	Proof-of-History (PoH)	Proof-of-Stake (PoS) Plasma-based sidechain	Proof-of-Staked Authority (PoSA)	Proof-of-Stake (PoS)
Transaction per second (TPS) [69]—Ethereum, Solana and Polygon	Anywhere from 13–25 TPS.	50,000–65,000 TPS.	65,000 TPS	Around 300 TPS [70]	Up to 250 TPS/75,000 Transaction per day (TPD)
Transaction confirmation time	After merging 12 s per Ethereum block.	1 s–400 s milliseconds	2 s–8 min	Up to 15 min	40 s
Transaction rate/Average Transition Fee (ATF)	$0.4729	$0.00025 per transaction	$0.018	$0.2214	$0.03–$0.47
Top NFT collections (high volume)	Ethereum Name Service (ENS), Board Ape Yacht Club, CryptoPunks, Otherdeed for Otherside	DeGods, Primates, Okaybears, Blocksmith Labs	Aavegotchi, PLAYZ Token V2 V2,	Mobox, PancakeSwap squad, Gamer NFLs	SapceBudz, Clay Nation, Pavia
NFT Market Places	OpenSea, Rarible, SuperRare	Solanart, Magic-Eden DigitalEyes, SolSea	AirNFTs, OpenSea, Refinable	Binance NFT, AirNFTs	CNFT.IO, JPG Store, Fibo
Security -Some of the known threats include: Phishing attacks 51% attacks Routing attacks Endpoint vulnerabilities Sybil attacks Denial of Service (DoS) etc.… [71]	It is highly secure due to its active community and active developers, as well as being one of the longest-running platforms.	Due to the security concerns associated with its small number of validators (roughly 1000), it is ranked fourth among secure blockchains.	Their Security comes in second to ETHEREUM due to its placement of the validation system and has similar security structures.	The blockchain is less secure because it is heavily centralized with 21 validators or miners, which makes it less robust and more vulnerable to manipulation. Nevertheless, they are still very secure blockchains.	The third most secure blockchain is due to its Ouroboros consensus mechanism which supports a provably safe and integrated community network. In addition, it also has a mathematical model that allows for the creation of provably secure blockchains.
Transparency	Public Blockchain (Permissionless access).	Public Blockchain (Permissionless access).	Public Blockchain (Permissionless access).	Public Blockchain (Permissionless access).	Public Blockchain (Permissionless access).
Scalability	The following were changed from layer 1 to layer 2: Staking, sharding, Off-chain scaling: Rollups (Optimistic rollups and Zero-Knowledge rollups), State channels, Sidechains, Plasma, and Validium. [72]	High-performance protocol scaling. Staking, Turbine, Tower BFT, Pipeline, Archives. [73]	Multichain Solutions offer to scale. Ethereum scaling Architecture (Layer 2), PoS chain, zero-knowledge Ethereum virtual machine (zkEVM), Plasma Sidechain, and more. [74]	Compatibility across chains. From a single BNB chain to a multichain chain. [75]	A second layer consists of Hydra Heads (state channels) operating on top of layer 1. Sidechains, and Mithril (stake-based threshold multi-signatures). [76]

**Table 4 sensors-22-08833-t004:** NFT marketplaces that are currently ranked among the top three.

Rank	NFT Marketplace	Active Users (2022)	Scalability (Gas and Wallet)	Sales Volume	Supported Blockchains	Security and Transparency	Best for?
**#1**	OpenSea	1,685,613 +	The marketplace is gas-free for mining but charges 2.5 for selling NFTs, and you can transfer NFTs to any wallet you like.	$2.64 B (past 30 days)	Ethereum, Polygon, Solana, Klatyn	Yes—User authentication Yes—Verification of tokens contract Yes—Seller/collection verification Yes—Transparency Yes—Royalty Yes—Fee evasion	Best overall
**#2**	Rarible	1.6 million users	A flow-based NFT, and a low gas fee are available. Wallets such as MetaMask, Coinbase wallet, Wallet Connect, MyEtherWallet, Torus, and Fortmatic are all supported.	$2.63 (24 h)	Ethereum	Yes—User authentication Yes—Verification of tokens contract Optional—Seller/collection verification Yes—Transparency Yes—Royalty Yes—Fee evasion	Best for Categories: digital art, collectibles, music, and video NFTs.
**#3**	SuperRare	7500 artists as of now	The marketplace has a fee of 3% on any transaction. They are currently working with only a few wallets, such as Meta mask, Fortmatic, and Wallet Connect.	31.4 million (monthly record)	Ethereum	Yes—User authentication Not appliable—Verification of tokens contract Mandatory—Seller/collection verification Yes—Transparency Yes—Royalty No—Fee evasion	Best for finding Artists: Art and music

## Data Availability

Not applicable.

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
