# Peer review of "A Survey: Security, Transparency, and Scalability Issues of NFT’s and Its Marketplaces"

_sensors, 2022, doi:10.3390/s22228833_

Round 1
Reviewer 1 Report
Summary : This is a survey paper on NFT marketplace and its dynamics. The article introduces the concept of NFTs and describes Blockchain as its base technology with specifying its pillars. The security aspects of NFT and discussions on NFT blockchain platforms and NFT Marketplace are carried out. Further, challenges faced by the NFT eco-systems are presented.
This is a well-written article.
Weak Aspects:
- This paper is not relevant to the special issue "Security and Privacy for Machine Learning Applications" and section on "Vehicular Sensing".
- Authors should relate their discussion towards the machine learning of vehicular applications. Maybe NFTs can produce a validation spectrum for such application areas.
- There are some articles which the authors have missed in compiling the section 2.
Chohan, Usman W. "Non-fungible tokens: Blockchains, scarcity, and value." Critical Blockchain Research Initiative (CBRI) Working Papers (2021).
Kugler, Logan. "Non-fungible tokens and the future of art." Communications of the ACM 64, no. 9 (2021): 19-20.
Valeonti, Foteini, Antonis Bikakis, Melissa Terras, Chris Speed, Andrew Hudson-Smith, and Konstantinos Chalkias. "Crypto collectibles, museum funding and OpenGLAM: challenges, opportunities and the potential of Non-Fungible Tokens (NFTs)." Applied Sciences 11, no. 21 (2021): 9931.
Chohan, Raeesah, and Jeannette Paschen. "What marketers need to know about non-fungible tokens (NFTs)." Business Horizons (2021).
Bao, Hong, and David Roubaud. "Non-Fungible Token: A Systematic Review and Research Agenda." Journal of Risk and Financial Management 15, no. 5 (2022): 215.
Eventhough they are not survey papers, their content and research directions are worth mentioning.
- The paper lacks technical analysis overall. All the sections should be improved with better technical analysis.
- Each security aspect/solution should be presented in terms of the cost it is incurring to the marketplace dynamics.
- Even for a survey paper, presented facts are quite vague. In comparisons, specifying certain aspects as slow/fast or costly/not costly is not scientific. Better to benchmark such qualitative measurements at least.
- Citations should be put with a standard way, some are mentioned in simple letters, while some are in capital.
- The topic of the Figure 1 : SOLD ==> Sold. Include a start or end date of this data representation to get a context on time.
- Include a figure for specifying the pillars of NFT.
- Include a figure for specifying the layer or architecture of the Blockchain.
Author Response
A revised version of the point-by-point response can be found in the attachment.
- There are some articles which the authors have missed in compiling the section. Even though they are not survey papers, their content and research directions are worth mentioning:
Bao, Hong, and David Roubaud. "Non-Fungible Token: A Systematic Review and Research Agenda." Journal of Risk and Financial Management 15, no. 5 (2022): 215.
Chohan, Raeesah, and Jeannette Paschen. "What marketers need to know about non-fungible tokens (NFTs)." Business Horizons (2021).
Valeonti, Foteini, Antonis Bikakis, Melissa Terras, Chris Speed, Andrew Hudson-Smith, and Konstantinos Chalkias. "Crypto collectibles, museum funding and OpenGLAM: challenges, opportunities and the potential of Non-Fungible Tokens (NFTs)." Applied Sciences 11, no. 21 (2021): 9931.
Kugler, Logan. "Non-fungible tokens and the future of art." Communications of the ACM 64, no. 9 (2021): 19-20.
Chohan, Usman W. "Non-fungible tokens: Blockchains, scarcity, and value." Critical Blockchain Research Initiative (CBRI) Working Papers (2021).
The article you provided has been mentioned
- The paper lacks technical analysis overall. All the sections should be improved with better technical analysis.
Technical analysis has been provided for all sections
2. Even for a survey paper, presented facts are quite vague. In comparisons, specifying certain aspects as slow/fast or costly/not costly is not scientific. Better to benchmark such qualitative measurements at least.
Detailed measurements were provided for slow/fast, and cost/no cost.
3. Citations should be put with a standard way, some are mentioned in simple letters, while some are in capital.
Citations follow a standard format.
4. The topic of the Figure 1: SOLD ==> Sold. Include a start or end date of this data representation to get a context on time.
A date has been input into figure 2 data, along with the change in title.
5. Include a figure for specifying the pillars of NFT and Include a figure for specifying the layer or architecture of the Blockchain.
Mention of the NFT workflow system derived from a study by Qin Wang titled " Non-fungible tokens (NFTs): Overview, Evaluation, Opportunities, and Challenges".
Implemented blockchain architecture

Reviewer 2 Report
The manuscript gives a survey about the security, transparency, and security issues of NFT and the marketplaces. The manuscript is well-organized, and provides an in-depth knowledge of underlying technologies and issues of NFT for the readers.
Several suggestions are listed in the following for the authors to improve the manuscript.
1. The Ethereum merge is done. Please revise the manuscript due to the latest update.
2. The performance (transaction per second and transaction per day) should be verified again. Please refer to the detailed suggestions in the following.
a. If the information is based on public benchmark or studies, please add the reference for readers.
b. The authors could elaborate on the performance in more detail. Is it the status quo, in theory, or the anticipated performance in the future? The information should be verified for readers.
3. Could the authors give their insights into the solutions to the security, transparency, and scalability issues?
4. Please proofread the manuscript for better readability.
Author Response
A revised version of the Point-by-point response can be found in the attachment.
1. The Ethereum merge is done. Please revise the manuscript due to the latest update.
Merge of Ethereum revised.
2. The performance (transaction per second and transaction per day) should be verified again. Please refer to the detailed suggestions in the following.
a. If the information is based on public benchmark or studies, please add the reference for readers.
References based on public benchmarks have been added.
b. The authors could elaborate on the performance in more detail. Is it the status quo, in theory, or the anticipated performance in the future? The information should be verified for readers.
All performance details (TSP) have been verified with additional information.
3. Could the authors give their insights into the solutions to the security, transparency, and scalability issues?
Outlining the solution for each section: security, transparency, and scalability.
4. Please proofread the manuscript for better readability.
Improved readability.

Round 2
Reviewer 1 Report
Reviewer would like to appreciate the authors rapid response to my comments. Still, there are further improvements required to raise the standards of this manuscript.
First of all, since the paper is being submitted to the Intelligent Sensors section, please provide a context in which NFTs can be relatable for sensory applications that are envisaged in the literature. Marketplace integration can be commented as well.
- Figure 1: Internal storage instead of Interal storage. No need to mention the name of the referred paper in the in the caption.
- Figure 2: Include the time duration in the figure title as well.
- Figure 3 does not explain the layers specified in section 3.4. Page 5 - figure 3 ==> Figure 3.
- Include references to backup the claims in Table 1.
- Still, there are citations that deviate from the common standard as in (RADOSŁAW MICHALSKI 2017). Please fix them.
- Organize Table 2 into a same page.
- 3.6.1 : facto ==> factor
- 3.6 Security section still can be extended to explaining the unpopular security issues such as exposure of the private key, masquerading NFT issuers, etc.
- Table 3: Use either TPS or tps. This table presents one of the important contrasts of the findings of this research. However, a comparison has not being conducted for security, transperencey, and scalability aspects. Please provide this comparison as well. Include a row under security for probable known security threats.
- Table 4 lacks the comparison on security, ....etc. Please extend the table accordingly.
- 6. Challenges faced by NFT Ecosystems
- Page 19 : fishing ==> Phishing
- Section 6.1: change the sub-topic Solutions to Possible Solutions. Most of the specified solutions are more generic. And there are more sophisticated approaches to securing systems in current literature. This discussion lacks any citations to state-of-the-art literature. Please update the possible solutions to the security issues.
- Update the possible solutions of sections 6.2 and 6.3 as well.
- Authors can add Metaverse as a future direction for NFT deployment surely.
Author Response
"Please see the attachment"

Reviewer 2 Report
The manuscript is highly improved. I have no further concerns about the manuscript.
Author Response
I appreciate you taking the time to provide feedback about the manuscript and appreciate the improvements that have been made.
Many thanks,
Sangam